# Id1 and PD-1 Combined Blockade Impairs Tumor Growth and Survival of *KRAS*-mutant Lung Cancer by Stimulating PD-L1 Expression and Tumor Infiltrating CD8^+^ T Cells

**DOI:** 10.3390/cancers12113169

**Published:** 2020-10-28

**Authors:** Iosune Baraibar, Marta Roman, María Rodríguez-Remírez, Inés López, Anna Vilalta, Elisabeth Guruceaga, Margarita Ecay, María Collantes, Teresa Lozano, Diego Alignani, Ander Puyalto, Ana Oliver, Sergio Ortiz-Espinosa, Haritz Moreno, María Torregrosa, Christian Rolfo, Christian Caglevic, David García-Ros, María Villalba-Esparza, Carlos De Andrea, Silvestre Vicent, Rubén Pío, Juan José Lasarte, Alfonso Calvo, Daniel Ajona, Ignacio Gil-Bazo

**Affiliations:** 1Department of Oncology, Clínica Universidad de Navarra, 31008 Pamplona, Spain; ibaraibar@vhio.net (I.B.); mromanm@unav.es (M.R.); mrremirez@unav.es (M.R.-R.); avilaltal@unav.es (A.V.); apuyaltocal@alumni.unav.es (A.P.); aoliver.5@alumni.unav.es (A.O.); mtorregrosa@alumni.unav.es (M.T.); 2Program in Solid Tumors, Center for Applied Medical Research, Program in Solid Tumors, 31008 Pamplona, Spain; milopez@unav.es (I.L.); mecay@unav.es (M.E.); mcollant@unav.es (M.C.); sortiz.9@unav.es (S.O.-E.); hmmoreno@unav.es (H.M.); silvevicent@unav.es (S.V.); rpio@unav.es (R.P.); acalvo@unav.es (A.C.); dajonama@unav.es (D.A.); 3Bioinformatics Platform, Center for Applied Medical Research, 31008 Pamplona, Spain; eguruce@unav.es; 4Program of Immunology and Immunotherapy, Center for Applied Medical Research, Program of Immunology and Immunotherapy, 31008 Pamplona, Spain; tlmoreda@unav.es (T.L.); jjlasarte@unav.es (J.J.L.); 5Flow cytometry Core Facility, Center for Applied Medical Research, Flow Cytometry Core Facility, 31008 Pamplona, Spain; dalignani@unav.es; 6Centro de Investigación Biomédica en Red de Cáncer (CIBERONC), 31008 Madrid, Spain; 7Thoracic Medical Oncology Program, Marlene and Stewart Greenebaum Comprehensive Cancer Center, University of Maryland, Baltimore, MD 21201, USA; christian.rolfo@umm.edu; 8Cancer Research Department, Fundación Arturo Lopez Perez, Santiago de Chile 7500921, Chile; oncodemia@yahoo.com; 9Department of Pathology, Anatomy and Physiology, University of Navarra, 31008 Pamplona, Spain; dagarc@unav.es (D.G.-R.); mvillalba.1@alumni.unav.es (M.V.-E.); ceandrea@unav.es (C.D.A.); 10IdiSNA, Navarra Institute for Health Research, 31008 Pamplona, Spain; 11Department of Biochemistry and Genetics, School of Sciences, University of Navarra, 31008 Pamplona, Spain

**Keywords:** KRAS lung adenocarcinoma, inhibitor of differentiation, PD-1 inhibition, PD-L1

## Abstract

**Simple Summary:**

Lung adenocarcinoma is the most frequent lung cancer subtype. Many of those adenocarcinomas of the lung are driven by the *KRAS* gene. Although immunotherapy has significantly improved the clinical outcomes of patients with lung adenocarcinomas, many patients do not benefit from that therapeutic strategy. Id1 is a protein involved in immunosuppression. Here we aimed to test whether a combined blockade of Id1 and PD-1 is able to improve outcomes of mice models with *KRAS*-driven lung adenocarcinoma.

**Abstract:**

The use of PD-1/PD-L1 checkpoint inhibitors in advanced NSCLC is associated with longer survival. However, many patients do not benefit from PD-1/PD-L1 blockade, largely because of immunosuppression. New immunotherapy-based combinations are under investigation in an attempt to improve outcomes. *Id1* (inhibitor of differentiation 1) is involved in immunosuppression. In this study, we explored the potential synergistic effect of the combination of *Id1* inhibition and pharmacological PD-L1 blockade in three different syngeneic murine *KRAS*-mutant lung adenocarcinoma models. TCGA analysis demonstrated a negative and statistically significant correlation between PD-L1 and *Id1* expression levels. This observation was confirmed in vitro in human and murine *KRAS*-driven lung cancer cell lines. In vivo experiments in *KRAS*-mutant syngeneic and metastatic murine lung adenocarcinoma models showed that the combined blockade targeting *Id1* and PD-1 was more effective than each treatment alone in terms of tumor growth impairment and overall survival improvement. Mechanistically, multiplex quantification of CD3^+^/CD4^+^/CD8^+^ T cells and flow cytometry analysis showed that combined therapy favors tumor infiltration by CD8^+^ T cells, whilst in vivo CD8^+^ T cell depletion led to tumor growth restoration. Co-culture assays using CD8^+^ cells and tumor cells showed that T cells present a higher antitumor effect when tumor cells lack *Id1* expression. These findings highlight that *Id1* blockade may contribute to a significant immune enhancement of antitumor efficacy of PD-1 inhibitors by increasing PD-L1 expression and harnessing tumor infiltration of CD8^+^ T lymphocytes.

## 1. Introduction

Primary lung cancer is the leading cause of cancer-related mortality [1]. Approximately 80% of all lung neoplasms correspond to non-small cell lung cancer (NSCLC), being lung adenocarcinoma (LUAD) the most frequent histology subtype [2].

The discovery of actionable driver genomic alterations that confer sensitivity to targeted therapies and the ability of immune checkpoint inhibitors (ICIs) to harness an anti-tumor immune response have revolutionized therapy for advanced NSCLC. The implementation of science-based precision medicine has resulted in a significant improvement in both patients’ life expectancy and quality of life [3]. However, the vast majority of LUADs either lack an identifiable driver oncogene or do not harbor *KRAS* mutations (25–30%) [4], and no pharmacological inhibitor for either of these circumstances has yet been approved for clinical use.

Anti-PD-1/PD-L1 monoclonal antibodies such as nivolumab, pembrolizumab, and atezolizumab have been widely investigated in metastatic NSCLC and have shown encouraging results as frontline therapy and in previously treated patients [5,6,7,8]. Nevertheless, only a small subset of patients obtain any long-term benefit from single agent immune checkpoint blockade and PD-L1 expression [9,10]. Combined strategies adding ICIs to chemotherapy regimens in NSCLC may improve antigen presentation to T cells and favor elimination of immunosuppressive elements from the tumor microenvironment, thus demonstrating a clinical synergistic anti-tumor effect [11]. Most clinical trials testing such combinations have shown efficacy in terms of overall survival (OS) and progression free survival (PFS) but at the expense of a higher rate of adverse events [12,13]. Recently, it has become apparent that cancer-targeted therapies, in addition to their anti-tumor activity, may potentiate T cell immune recognition of tumor cells, resulting in a potentially synergistic improvement of the efficacy of ICIs [14,15].

Inhibitor of differentiation (Id) genes (*Id1–4*) are a family of highly conserved transcriptional regulators that are critical in both developmental processes and adult tissue homeostasis. Id proteins lack the basic amino acids needed for DNA binding. They form non-functional heterodimers by binding to basic helix-loop-helix proteins (bHLH), ETS, and paired-box (PAX) transcription factors and non-transcription factors of the Rb family; thus, they act as dominant negative regulators, inhibiting their functions. Id proteins (mainly Id1 and Id3) have been demonstrated to be overexpressed in human cancers, including lung primary tumors [16,17]. Deregulation of Id proteins affects cancer initiation, maintenance, and progression [18] and also plays a role in suppressing the antitumor immune response during tumor progression and metastasis through downregulation of molecules involved in dendritic cell differentiation and suppression of CD8^+^ T-cell proliferation [19]. Id1 expression has been proven to correlate with poorer OS in LUAD and to be a marker of poor response specifically in the *KRAS*-mutant LUAD setting, providing the cells with a more aggressive pro-oncogenic phenotype [18,20]. In vivo experiments using murine LUAD models showed that Id1 expression in both lung cancer cells and the host microenvironment enabled liver metastasis by increasing migration and colonization, thereby favoring the establishment of a pre-metastatic niche by regulating some epithelial-to-mesenchymal transition (EMT) proteins [21]. In pancreatic ductal adenocarcinoma, dysregulation of *Id1* has been proved to counter the apoptotic effect of TGF-β by decoupling TGF-β-induced EMT from apoptosis [22]. Moreover, *Id1* plays a role in several immune system-related processes such as the differentiation of regulatory T cells (Treg) and the impairment of myeloid cell maturation [19,23]. However, the potential synergistic effect of the combination of *Id1* inhibition and PD-L1 blockade in *KRAS*-mutant LUAD had not been evaluated. Here we provide key evidence of the benefit of this combined blockade in *KRAS*-mutant murine LUAD models and characterize the mechanisms involved.

## 2. Results

### 2.1. Id1 Expression is Associated with PD-L1 Expression in LUAD Patients

The potential correlation in LUADs between *Id1* expression levels and the expression of several immune response markers consisting of a six-gene *IFN-γ* signature [24] (*CXCL9, CXCL10, IDO1, HLA-DRA, STAT1, IFN-γ*)*,* markers of immune cell populations (*FOXP3, CD3G, CD4, CD8A and CD68*), and immune checkpoints [*CD274 (PD-L1), PDCD1LG2 (PD-L2), CD80, CTLA4*] was first assessed using LUAD patient data available in the TCGA data set [25] (Table 1).

*p*-value and the magnitude of the correlation are shown in Table 1 and ranked in Figure 1A. No correlation was found between mRNA expression of *Id1* and the expression of *CD8a, CD68,* and *IFN-γ* (*p* > 0.05). Inverse and statistically significant correlations were found for the other immune response markers (*CXCL9, CXCL10, IDO1, HLA-DRA, STAT1, FOXP3, CD3G, CD4, PD-L1, PD-L2, CD80* and *CTLA4)* that were analyzed, suggesting that *Id1* may negatively regulate their expression. The top-ranked association with *Id1* was found for the expression of *PD-L1* (r = −0.35, *p* < 0.0001), suggesting that *Id1* may affect the therapeutic activity of PD-1/PD-L1 antagonists. Correlations for *CD4* and *HLA-DR* were −0.20 (*p* < 0.001) and −0.23 (*p* < 0.0001), respectively.

In view of the importance of *Id1* in the context of *KRAS* in LUAD [18], we explored whether the inverse correlation observed for *Id1* and *PD-L1* was dependent on the status of the *KRAS* oncogene. For this purpose, patients in the TCGA LUAD data set were stratified based on *KRAS* status (mutant and wild-type *KRAS*). No significant differences associated with *KRAS* mutational status were observed (Appendix A). However, a moderate and statistically significant correlation was found in both cohorts (r = −0.367 and *p* = 0.008 for mutant *KRAS* LUAD patients; r = −0.351 and *p* = 0.005 for wild-type *KRAS* LUAD patients). This finding suggests that the suppression of *Id1* may promote PD-L1 expression in LUAD tumor cells independently of *KRAS* status.

### 2.2. Up-Regulation of Surface PD-L1 Expression Occurs in Id1-Deficient KRAS Mutant LUAD Cells Exposed to IFN-γ

Previously, we found that *Id1* inhibition in both human H1792 and murine LLC cells was significantly associated with a significant reduction of cell proliferation in *Id1*-deficient LLC cells [21]. To assess the in vitro effect of silencing *Id1* in other murine LUAD cell lines, Lacun3, and 393P cells, *Id1* was knocked down using a constitutive shRNA against *Id1* (Id1sh) (Appendix A). A significant impairment in cell growth was observed in both cell lines upon *Id1* inhibition after 5 days in comparison with control cells (393P *p* = 0.001; Lacun.3: *p* = 0.0004) (Appendix A). To further characterize the cellular mechanisms underlying the effect of Id1 decreased protein expression, we assessed cell cycle arrestment in murine *KRAS*-mutant LUAD cell lines (LLC, Lacun3 and 393P). Cell cycle analyses using EdU incorporation flow cytometry assay were performed. *Id1* inhibition in these cells induced an arrest in G2/M phase in 393P (*p* = 0.0006) and LLC cells (*p* = 0.0161), whereas no significant changes were detected in Lacun3 cell line (Appendix A).

To evaluate the potential role of *Id1* in tumor-associated immune responses, we used flow cytometry to determine whether stimulation of a human *KRAS* mutant LUAD cell line (H1792) with IFN-γ, which regulates expression of immunomodulatory molecules on tumor cells [21,22,23], affected the expression of PD-L1 on the cell surface of such human tumor cells (Figure 1B,C). H1792 cells lacking *Id1* expression showed a significant increase in PD-L1 expression on the cell surface when they were previously stimulated with INF-γ over 24 hours (H1792 -IFN-γ: *p* = 0.0022). Moreover, we wanted to determine if the absence of the expression of *Id1* had the same effect on the expression of PD-L1 in murine *KRAS* mutant LUAD cells (LLC, Lacun3 and 393P) (Figure 2A). Flow cytometry analyses revealed that the lack of *Id1* resulted in the upregulation of PD-L1 expression on the cell surface, in some cases irrespectively of the presence of IFN-γ (LLC -IFN-γ: *p* = 0.005; 393P -IFN-γ: *p* = 0.0022). IFN-γ-mediated PD-L1 expression was significantly higher in 393P and Lacun3 *Id1*-depleted cells (393P +IFN-γ: *p* = 0.0022; Lacun3 +IFN-γ: *p* = 0.005). These in vitro data confirm our TCGA findings that showed that *Id1* plays a relevant role in the immunoregulatory mechanisms of the PD-1/PD-L1 axis both in human and murine models. In addition, we investigated whether the lack of *Id1* in these *KRAS* murine LUAD cell lines had also an effect in the membrane surface expression of PD-L2 and CD80. Flow cytometry showed no significant changes in these immune markers suggesting that PD-L1 is the most relevant immune marker in this specific context.

### 2.3. Combined Id1 Inhibition in Tumor Cells and PD-1 Blockade Impairs KRAS-Driven LUAD Growth In Vivo

To define the synergistic effect of *Id1* inhibition in tumor cells and PD-1/PD-L1 axis blockade in the in vivo setting, syngeneic models were generated by injection of our three *KRAS*-mutant murine LUAD cells (*Id1*sh-cells or pLKO-sc control cells) into the flanks of mice (*n* per group = 8). A significant decrease in tumor volume was observed in mice injected with *Id1*sh LLC cells that received anti-PD-1 agent compared with that seen in mice injected with *Id1* or PD-1 inhibition alone or that seen in the control group; and, after day 14, *p*-values for the groups were as follows: *Id1*sh/DPBS *p* = 0.0384; pLKOsc/anti-PD-1 *p* = 0.0012; pLKOsc/DPBS *p* = 0.0017 (Figure 2B,C).

Next, the effect of *Id1* inhibition in anti-PD-1 therapeutic effect was further validated in two additional *KRAS*-mutant syngeneic models: Balb/c mice receiving subcutaneous inoculation of Lacun3 cells and Sv/129 mice subcutaneously inoculated with 393P cells (*n* per group = 6). Animals injected with *Id1*-depleted Lacun3 cells and treated with anti-PD-1 agent experienced a significant tumor growth impairment (*Id1*sh/DPBS n.s; pLKOsc/anti-PD-1 *p* = 0.0037; pLKOsc/DPBS *p* = 0.0174) (Figure 2D,E). Taken together, these in vivo findings support the potential synergistic effect of the combined blockade in tumor growth impairment, compared to results seen with each inhibition strategy alone.

### 2.4. Combined Id1 Inhibition in Tumor Cells and PD-1 Blockade Impairs Tumor Progression in a Clinically Relevant Metastasis Model

To further investigate the impact of *Id1* inhibition on tumor progression, a murine model of cancer colonization to the liver was generated using LLC tumor cells (parental or *Id1* silenced; *n* per group = 5), as previously described [21]. A marked delay in liver metastasis formation was observed in the mice cohort inoculated with *Id1*sh-LLC and receiving treatment with anti-PD-1 monoclonal antibody in comparison to results seen in the other experimental groups. Of note, two mice from *Id1*/PD-1 blockade group did not present any kind of neoplastic lesions.

A representative mPET image from each group and final total MTV values for each mouse group are plotted in Figure 2F. Even though these differences were not statistically significant, there was a clear trend towards a delay in the liver tumor colonization in *Id1*-deficient LLC-tumor-bearing mice (*p* > 0.05). These data are in line with the idea that combined *Id1*-deficiency in *KRAS-*mutant murine LUAD cells enhances PD-1 blockade antitumor activity and leads to a lower propensity for liver colonization.

### 2.5. Id1 Inhibition in the Tumor Microenvironment Potentiates Response to Anti-PD-1 In Vivo and Improves Mouse Overall Survival

To investigate the potential synergistic impact of the PD-1 blockade and tumor microenvironment *Id1* blockade, *Id1^-/-^* (IDKO) and C57BL/6J (*Id1^+/+^*) mice were injected with parental LLC cells (*n* per group = 8). *Id1^-/-^* mice treated with the anti-PD-1 monoclonal antibody showed a significantly reduced tumor growth and longer survival in comparison to results seen in control groups (*Id1^-/-^*/DPBS *p* = 0.0283; *Id1^+/+^*/anti-PD-1 *p* = 0.0039; *Id1^+/+^*/DPBS *p* < 0.0001) (Figure 3A–C). Comparison between *Id1^-/-^* and *Id1^+/+^* strains revealed that animals deficient in *Id1* that were treated with anti-PD-1 agent showed a significantly increased survival (*p* = 0.0035). This model supports the synergistic effect of the combination of *Id1* loss in the tumor microenvironment and PD-1 blockade.

### 2.6. Combined Id1 Inhibition in both Tumor Cells and Tumor Microenvironment and PD-1/PD-L1 Blockade Impairs KRAS-Driven LUAD In Vivo Growth

Next, we studied the impact of the combined blockade when both the host and the tumor cells lacked *Id1* expression. The comparison between *Id1^-/-^* and *Id1^+/+^* LLC *Id1*sh tumor-bearing mice (*n* per group = 8) showed that the lack of *Id1* in both tumor cells and tumor microenvironment significantly potentiated the antitumor activity of anti-PD-1 therapy (*Id1^-/-^*/DPBS *p* = 0.0207; *Id1^+/+^*/anti-PD-1 *p* = 0.0269; *Id1^+/+^*/DPBS *p* = 0.0017) (Figure 3D,E). 

Taken together, these experiments suggest that the lack of Id1 expression in both tumor cells and the tumor microenvironment, along with the blockade of PD-1/PD-L1 axis, exerts the most potent antitumor effect in murine *KRAS-*driven LUAD (Figure 3F).

### 2.7. Combined Inhibition of Id1 and PD-1 Increases the Frequency of Effector T Cells within the Tumor Microenvironment

To determine the mechanisms involved in the antitumor activity observed in response to the *Id1-*loss in combination with the blockade of PD-1/PD-L1 axis, we evaluated the changes induced in the tumor microenvironment by the combined blockade. We performed an immunohistochemical analysis of tumor infiltrating lymphocytes (TIL) from *Id1^+/+^*/DPBS (pLKO-sc LLC), *Id1^+/+^*/anti-PD1 (pLKO-sc LLC), *Id1^-/-^/*DPBS (*Id1*sh LLC), and *Id1^-/-^/*anti-PD1 (*Id1*sh LLC) tumors (Table 2 and Figure 4). Interestingly a significant increase in TIL was observed specifically in mice with the combined inhibition of *Id1* in both tumor cells and tumor microenvironment and PD-1 (Figure 4A). Isolated inhibition of *Id1* in both tumor and host cells enhanced the frequency of tumor-infiltrating CD8^+^ T cells. However, the analysis of the number of CD8^+^ (Figure 4B) and CD4^+^ T cells (Figure 4C) showed the highest tumor infiltration by both types of lymphocytes in *Id1* knockout mice bearing *Id1*-silenced tumor cells treated with anti-PD-1 therapy. 

Within the population of CD4^+^ cells, we found no significant changes associated with the double blockade of *Id1* and PD-1 in the percentage of Treg cells stained with the FOXP3 marker (Figure 4D). Additionally, quantitative multiplex IHC (CD3, CD4 and CD8) revealed an increase in the percentage of CD3^+^ TILs and a clear tendency towards high infiltration of CD8^+^ lymphocytes in tumors derived from *Id1* knockout mice injected with *Id1*sh-LLC cells and treated with anti-PD-1 therapy. 

However, in this case, no significant changes related to CD4^+^ lymphocytes were observed (Figure 5A,B). These results indicate that lack of *Id1* in both tumor cells and tumor microenvironment, along with the blockade of the PD-1/PD-L1 axis favors the infiltration of different immune cell populations into the tumor, specifically the presence of CD8^+^ T lymphocytes, which may justify the antitumor activity associated with the double blockade.

### 2.8. The Antitumor Activity of the Combined Blockade May Be Mediated by CD8^+^ T Cells

Based on these results, we decided to further investigate the potential relationship between *Id1* and the recruitment of CD8^+^ T cells into the tumor. The first step was to conduct a co-culture assay in LLC-GFP-ovalbumin (OVA) (LLC TMG) and LLC-GFP tumor cells (pLKO-sc or *Id1*sh) with OT-I CD8^+^ T cells. After 24 hours of co-culture, the capacity of OT-I CD8^+^ T cells to recognize and lyse tumor cells expressing OVA was analyzed with flow cytometry. Interestingly, when the E:T ratio was 1:1, OT-I CD8^+^ T cells were clearly more efficient at killing *Id1*sh-LLC TMG cells with respect to pLKO-LLC TMG cells (*p* = 0.0024) (Appendix A). These differences suggest that the lack of *Id1* in murine *KRAS*-mutant LLC cells facilitates the activation of CD8^+^ T lymphocytes, which may account for the anti-tumor response derived from *Id1* inhibition.

To better understand the molecular mechanisms underlying in the *Id1-*loss phenotype in combination with the blockade of the PD-1/PD-L1 axis, we investigated the potential role of each immune cell subpopulation in the observed antitumor effect on the tumor growth when both *Id1* and PD-1 were inhibited. C57BL/6J mice were inoculated with *Id1sh*-LLC cells (*n* per group = 8) and treated with depleting antibodies against CD8, NK, or CD4 cells in combination with the anti-PD-1 therapy. We observed that a selective depletion of CD8^+^ T cells completely abrogated the antitumor efficacy of the *Id1*-PD-1 double blockade against LLC cells and restored the expected tumor growth (Id1sh/DPBS, n.s.; Id1sh/anti-PD-1, *p* = 0.0425) (Figure 5C,D). In contrast, while NK cells seemed to play no significant role in the anti-tumor activity (Appendix A), CD4^+^ T cells functional blockade caused a drastic decrease in tumor growth that could be attributed to CD4^+^ Treg cell depletion (Appendix A).

Finally, we characterized the intratumor immune cell subpopulations to corroborate if CD8^+^ T cells indeed had an active and relevant antitumor role in murine *KRAS-*mutant LUAD. We treated established LLC tumor-bearing mice (*n* per group = 8; extreme groups: *Id1^+/+^*/DPBS injected with pLKO-sc LLC; *Id1^+/+^*/anti-PD-1 injected with pLKO-sc LLC; *Id1^-/-^*/DPBS injected with *Id1sh*-LLC; *Id1^-/-^*/anti-PD-1 injected with *Id1sh*-LLC) with anti-PD1 therapy or vehicle during a defined period of time.

At day 14 after treatment initiation, we observed a very significant decrease in tumor growth in *Id1* knockout mice treated with an anti-PD1 monoclonal antibody (*Id1^+/+^*/anti-PD-1, *p* = 0.0029; *Id1^+/+^*/DPBS, *p* < 0.0001) (Figure 6A). The animals were sacrificed at that point, and the tumors were processed to characterize the presence of several immune cell subpopulations by flow cytometry. As in previous results, we observed an increase in the proportion of intratumor CD8^+^ T cells (Figure 6B); whereas, no significant changes were seen in the frequency of tumor-infiltrating CD4^+^ T cells (Figure 6C). Moreover, the most significant upregulation was associated with the population of effector CD8^+^ T cells (Appendix A). We also analyzed the expression of surface activation markers in tumor-infiltrating CD8^+^ T cells. These cells showed a marked reduction in the exhaustion marker PD-1, which was already downregulated in those tumors treated with anti-PD-1 alone. A clear tendency towards a higher expression of GITR and LAG3 markers was also observed for tumor-infiltrating CD8^+^ T cells (Figure 6D). No relevant changes in the frequencies of NK and B lymphocytes were observed among the different experimental groups ( Appendix A). Furthermore, in the myeloid cell populations, we found a decrease in the frequency of MDSC leucocyte subpopulation in tumors in *Id1^−/−^* mice, as compared with results seen in *Id1^+/+^* animals (Figure 6E and Appendix A). In contrast, none of the treatment arms significantly modified the frequency of dendritic cells (Appendix A). Considered together, these findings suggest that CD8^+^ lymphocytes may be the main cell immune subpopulation mediating the antitumor activity when *Id1sh*-LLC tumor-bearing mice are treated with an anti-PD-1 antibody.

## 3. Discussion

Despite the notable clinical improvements derived from the use of ICIs in lung cancer therapy, a significant proportion of patients do not benefit from these treatments [26]. Intensive investigation is being carried out to learn more about the potential mechanisms of resistance and possible strategies to overcome it. These studies are based primarily on PD-1/PD-L1 axis inhibition combined with other drugs against a number of immune system targets [27]. In this context, a better understanding of the factors that mediate antitumor response to ICIs is at the core of the development of rational combinations with other therapies.

Our previous work has revealed that the expression of the dominant negative transcriptional regulator *Id1* is a biomarker of poorer prognosis as it is associated with tumor progression, metastasis formation, and tumor-immunosuppression [23,28] in *KRAS*-driven NSCLC [21]. Here we describe, for the first time, the synergistic effect of the combined blockade of *Id1* and PD-1 in *KRAS*-driven LUAD murine models and the immune mechanisms that may be involved, confirming previous studies showing the immunosuppressive role of *Id1*. The rationale behind this approach is that combined treatment may partially reverse the immunosuppressive tumor microenvironment, favor the immune infiltration of tumors, and thereby exert a relevant antitumor activity.

In this unique experimental approach, we first found in silico that in human cohorts included in TCGA, *Id1* expression inversely correlates with the mRNA expression levels of several markers related to immune response and that the suppression of *Id1* promotes PD-L1 expression on the tumor cell surface irrespective of *KRAS* status. Interestingly enough, *Id1* inhibition in vitro induced G2/M phase cell cycle arrest in murine *KRAS*-mutant LUAD cell lines, recapitulating the results previously observed in *KRAS*-driven LUAD cells of human origin [18]. Moreover, Id1 silencing also resulted in significant increased levels of surface *PD-L1* expression in the murine *KRAS*-driven lung cancer LLC, Lacun3 and 393 P cell lines, often even without the need for a pro-inflammatory stimulus (IFN-γ exposure in this case). The mechanism by which *Id1* induces the upregulation of *PD-L1* needs clarification; it could be due to the fact that *Id1* regulates the expression of several targets, including *TNK2*, *GRK6* or *RSK-B* kinases, or *ERK1/2* and *FOSL1* regulation, establishing a crosstalk within the MAPK signaling pathway [18]. MAPK cascade might play a role in *Id1′*s immunosuppressive function as a result of its relationship to apoptosis inhibition, cell survival, MHC expression impairment [29], and T-cell function inhibition [30].

Our in vivo experiments in which *Id1* was silenced in tumor cells, host cells, or in both revealed the important role of *Id1* expression in tumor and host cells in the antitumor activation of the immune system. Combining Id1 inhibition with anti-PD-1 treatment produced a clear synergistic effect that was more effective than either strategy alone in terms of tumor growth impairment and overall survival. These results were reproducible and consistent in different murine models, supporting our hypothesis. The most plausible biological mechanism behind this finding is that combined *Id1*/PD-1 inhibition favors tumor infiltration by activated immune cells. Moreover, co-culture assays demonstrated that *Id1* deficiency in tumor cells increased the lethality of CD8^+^ T cells. We have shown by both multiplex quantification and flow cytometry that activated CD8^+^ T lymphocytes infiltrating the tumor are the main represented immune cell population. More interestingly, CD8^+^ T cell depletion led to tumor growth phenotype restoration due to a significant reduction in the efficacy of PD-1 blockade, despite inhibition of *Id1* and anti-PD-1 monoclonal antibody administration. These results indicate that CD8^+^ T cells are the main population involved in exerting the antitumor response. Although NK cell depletion did not interfere with the antitumor activity of the combination therapy, a delay in the tumor growth was observed when antibodies depleting CD4^+^ T cells were administered. Pleiotropic functions of CD4^+^ T cells in antitumor response have been described, including effector cytokine secretion, recruitment of effector cells, involvement as helpers of CD8^+^ T cells for priming and effector function, generation of CD8^+^ memory T cells, and CD4^+^ cells acting as T regulatory cells [31,32,33,34,35,36]. In fact, the impairment in tumor growth after CD4^+^ T cell depletion could be mediated by the elimination of the CD4^+^ Treg cells. Higher expression of the GITR and LAG3 markers in CD8^+^ T cells shown by flow cytometry may be explained by the upregulation of GITR expression upon T cell activation, as previously reported [37,38].

In addition to our findings in the study of the *Id* genes in lung cancer, other groups have demonstrated the role of the *Id* genes as negative regulators of the immune system. Moreover, it is known that *Id1* is involved in the development, maturation, and differentiation of thymocytes into CD4^+^ and CD8^+^ T cells. Therefore, the overexpression of *Id1* produces a decrease in these populations [39]. Liu et al. demonstrated that the number of regulatory FOXP3^+^ T cells is duplicated in the thymus of *CD4-Id1* transgenic mice, with similar results in vitro [23]. Likewise, other researchers have proposed that the intervention of *Id1* and *Id3* in the maturation process of dendritic cells derived from monocytes is induced by bone morphogenetic proteins (BMPs) and that the stimulation of BMP-4 induces the expression of PD-L1 and PD-L2 [40].

A recent study with a novel BMP receptor inhibitor, JL5, has shown the suppression of the expression of *Id1* in human lung cancer xenografts. Moreover, the authors demonstrated JL5-induced tumor cell death and tumor regression in xenograft mouse models without immune cells and humanized with adoptively transferred human immune cells. Interestingly enough, in humanized mice, the BMP receptor inhibitor additionally induced the infiltration of immune cells within the tumor microenvironment [41]. Another study by a group at Weill Cornell Medical College [19] showed that *Id1* expression induced a defect in the maturation of myeloid cells during tumor progression in mice implanted with B16F10 melanoma cells, and they observed six times more metastases after transplantation with chimeric bone marrow *Id1*^+/+^ [19]. In an analysis performed in patients with stages III and IV melanoma, a high *Id1* expression significantly correlated with the expression of phenotypic and immunosuppressive markers of human monocytic myeloid-derived suppressor cells, and *Id1* downregulation in healthy donors led to differentiation from the monocytic lineage to a more immunogenic phenotype [42]. Additionally, it is also known that the *Id* family participates in the development of NK cells [43].

More recently, *Id1* has been shown to repress *TGF-β*-induced apoptosis signaling in pancreatic cancer. While *Id1* downregulation was deleterious to cells derived from pancreatic tumors that retained functional *TGF-β* pathway, they were protected from apoptosis when *Id1* was dysregulated and *SOX4* expression was decreased [22].

Combined blockade of *Id1* and PD-1 could also synergize through an antiangiogenic effect, as *Id1* is part of the cascade that mediates VEGF pro-angiogenic effects [44,45]. Loss of *Id1* expression might have a negative effect in neovascularization and restore leukocyte adhesion of immune cells trafficking into the tumor microenvironment that is defective in neovessels and thereby enhance the effect of PD-1/PD-L1 axis blockade [46].

More recently, a small-molecule *Id*-antagonist, AGX51, has been developed and has proven to be effective in ubiquitin-mediated degradation of Id proteins, cell growth arrest, and reduced cell viability [47]. In further research, it would be interesting to study the effect of AGX51 in combination with an anti-PD-1/PD-L1 agent. Treatment with AGX51 has been reported to be safe in murine models [47], but the potential toxicity associated with the combined treatment should also be assessed.

The present work has some limitations, as experiments have only been performed in immunologically competent murine models; and, although *Id1* inhibitors are under development, pharmacological inhibition of *Id1* is not yet available in clinical practice. However, the results from TCGA analysis and experiments in human and murine *KRAS*-driven LUAD cell lines here presented, in addition to the potential applicability of the combined blockade, should be expected to improve immunotherapy-based treatment outcomes, particularly as these results are confirmed in humanized murine models.

## 4. Materials and Methods

### 4.1. Id1-PD-L1 Correlation

The Cancer Genome Atlas (TCGA) data for the lung adenocarcinoma RNA-Seq experiments were downloaded from Genomics Data Commons (GDC) Data Portal (https://portal.gdc.cancer.gov) and analyzed in R/Bioconductor [48]. First, gene counts were normalized with edgeR [49] and voom [50]. Then, a Pearson correlation analysis in tumor samples was performed between *Id1* and several markers related to immune response: CD274 (PD-L1), PDCD1LG2 (PD-L2), CD80, CXCL9, CXCL10, IDO1, HLA-DRA, STAT1, CTLA4, FOXP3, IFN-γ, CD3G, CD4, CD8A, and CD68.

### 4.2. Cell Line Cultures

Human LUAD cell line H1792 and murine LUAD cell line Lewis lung carcinoma (LLC) were obtained from the American Type Culture Collection (ATCC, Manassas, VA, USA). The 393P cells, derived from *Kras^LA1/+^*; *p53^R172HΔG^* mice, were a gift of Jonathan M. Kurie (The University of Texas, MD Anderson Cancer Center, Houston, TX, USA). The Lacun3 cell line, which was established from a chemically induced lung adenocarcinoma [51], was a gift of Luis Montuenga (Center for Applied Medical Research; CIMA, Pamplona, Spain). Murine LUAD cells were cultured in RPMI 1640 medium (Gibco, Waltham, MA, USA) supplemented with 10% HyClone FetalClone II (Thermo, Waltham, MA, USA), 1% Penicillin/Streptomycin (Gibco), and HEPES (Lonza, Basel, Switzerland). All cell lines were routinely tested for Mycoplasma using a MycoAlert Mycoplasma Detection Kit (Lonza). Cell line authentication was not routinely performed.

The Platinum Ecotropic cell line (Plat-E, ATCC) was cultured in DMEM (Gibco) supplemented with 10% fetal calf serum (FCS, Sigma-Aldrich, Saint Louis, MO, USA) and the selection antibiotics puromycin (100 ug/ml) and blasticidin (10 μg/mL, Sigma-Aldrich).

### 4.3. Silencing Id1 Expression

Oligonucleotides for constitutive *Id1* shRNA (*Id1*sh) (TRCN0000071436) were: forward 5’-CCG GGCGAGGTGGTACTTGGTCTGTCTCGAGACAGACCAAGTACCACCTCGCTTTTTG-3’: reverse 5’-AATTCAAAAAGCGAGGTGGTACTTGGTCTGTCTCGAGACAGACCAAGTACCACCTCGC-3’ (Thermo, Waltham, MA, USA). Oligonucleotides were annealed and cloned into the lentiviral plasmid pLKO.1 (plasmid #8453, Addgene). pLKO.1-scramble (pLKO-Sc, plasmid #1864, Addgene) was used as control. Oligonucleotides for doxycycline-inducible *Id1* shRNA (i-*Id1*sh) (TRCN0000019029) were: forward 5′-CCGGCCTACTAGTCACCAGAGACTTCTCGAGAAGTCTCT GGTGACTAGTAGGTTTTTG-3′; reverse 5′-AATTCAAAAACCTACTAGTCACCAGAGACTTCTC GAGAAGTCTCTGGTGACTAGTAGG-3′ (Sigma-Aldrich). An inducible GFP shRNA (i-GFPsh) was used as control. Oligonucleotides were annealed and cloned into the lentiviral plasmid pLKO-Tet-On (plasmid #21915, Addgene). The Mission lentiviral packaging system (Sigma-Aldrich) was used to generate the lentiviral particles, and cells were infected as previously described^21^. Selection of transfected clones was carried out with puromycin (2 µg/mL).

### 4.4. Plasmid and Retroviral Production

Plasmid expressing SIINFEKL epitope was prepared by subcloning it into the KMV IRES GFP plasmid. The empty vector was used as a control. The Plat-E cell line was used for retrovirus production (RV-SIINFEKL or RV-GFP). Packaging cells were transfected with 5 μg of retroviral plasmid (KMV-IRES-GFP or KMV-IRES-SIINFEKL-GFP) and 2.5 μg pCL-Eco plasmid DNA using lipofectamine (Thermo Fisher Scientific). Retroviral supernatants were collected at 48 and 72 h. LLC cells transduced in the presence of 10 μg/mL protamine sulfate (Sigma). Infection was repeated on consecutive days. After one week, transduced LLC were sorted using GFP expression by Area-Sorter (BD, San José, CA, USA).

### 4.5. Western Blot Analyses

For Western blot analyses samples were processed in RIPA lysis buffer, and protein quantification was determined with the BCA Protein Assay Kit (Pierce, Waltham, MA, USA). Membranes were blocked with 5% nonfat dry milk in DPBS-Tween and incubated with antibodies raised against Id1 (1:2500 Biocheck, San Francisco, CA, USA) or GAPDH (1:5000, Abcam, Cambridge, United Kingdom). Membranes were incubated with the peroxidase substrate Lumi-Light Plus (Roche) and developed using a ChemiDoc Imaging System (BioRad, Hercules, CA, USA).

### 4.6. Cell Proliferation

Cells were seeded in 96-well plates in 100 µL of complete medium (500 cells per well for LLC and 393P cell lines; and 700 cells per well for Lacun3) in triplicate for each experimental condition. Cell proliferation was assessed at different time points (days 0, 3, and 5) by adding 10 μl of MTS reagent (Promega, Madison, WI, USA) per well and incubating for 2 hours at 37 °C. Absorbance was measured at 490 and 650 nm in a SPECTROstar Nano (BGM LABTECH, Ortenberg, Germany).

### 4.7. Cell Cycle Assay

Cell cycle analysis was performed using the Click-iT EdU Flow Cytometry Assay Kit (Invitrogen, Carlsbad, CA, USA). Cells were seeded and maintained in culture for 24h. Then, cells were incubated with 10 μM 5-ethynyl-2′-deoxyuridine (EdU) for 2 h, harvested, washed in DPBS containing 1% BSA and fixed in formaldehyde (Click-iT fixative) for 15 min at room temperature. Cells were washed in DPBS containing 1% BSA to remove formaldehyde, and permeabilized in 1× Click-iT saponin-based permeabilization and wash reagent for 15 min at room temperature. Next, cells were incubated for 30 min at room temperature in the dark, with the Click-iT reaction cocktail. After a washing step with 1× Click-iT saponin-based permeabilization and wash reagent, cells were incubated with 0.2 μg*μl^−1^ RNase A (Sigma-Aldrich) for 1 h at room temperature, in the dark. 7AAD was added to the tubes 10 min before the acquisition of cells in a FACSCanto II cytometer (BD Biosciences). Data were analyzed using FlowJo software v9.3 (BD, San José, CA, USA).

### 4.8. Co-Culture Assay

CD8^+^ T cells were isolated from OT-I mice spleens by magnetic selection (Miltenyi) and cultured with LLC-SIINFEKL *Id1*sh or LLC-SIINFEKL pLKO-sc at different ratios CD8^+^: target cells (5:1; 1:1 and 0.2:1). LLC-GFP *Id1*sh or LLC-GFP pLKO-sc were also used. Twenty four hours later, cells were harvested to measure the numbers of tumor cells by flow cytometry with Perfect Count Microspheres (Citognos, Salamanca, Spain). The percentage of specific lysis for each well was calculated as follows: % specific lysis = 100–(numbers of LLC-SIINFEKL in the presence of CD8^+^ T cells/mean of the three wells containing target cells alone)/(numbers of LLC-GFP in the presence of CD8^+^ T cells/mean of the three wells containing target cells alone) × 100.

### 4.9. Murine Models

All animal procedures were approved by the institutional Committee on Animal Research and Ethics (regional Government of Navarra) under the protocol numbers CEEA E51-16(113-13E2), 057-18 and 049-18. This study included 8-week-old female C57BL/6J mice, BALB/c mice (The Jackson Laboratory, Bar Harbor, ME, USA), Sv/129 mice (Janvier Labs, Le Genest-Saint-Isle, France), and *Id1*-deficient (*Id1^−/−^* or IDKO) mice with C57BL/6J background. *Id1^−/−^* mice were kindly provided by Robert Benezra (Memorial Sloan-Kettering Cancer Center, New York, NY, USA). OT-I transgenic mice (C57BL/6-Tg [Tcra/Tcrb] 1100 Mjb/J) with a TCR recognizing H2-Kb-restricted OVA (257–264 SIINFEKL peptide) were obtained from Jackson Laboratory.

Murine LLC, 393P, or Lacun3 cells (1.5 × 10^6^) infected with *Id1*sh or pLKO-Sc were injected subcutaneously in the right flanks of 8-week-old C57BL/6J (*Id1*^+/+^ and *Id1^−/−^*), Sv/129, or BALB/c mice. Tumor-bearing mice were treated with DPBS or 100 μg anti-PD-1 blocking antibody (RMP1-14, BioXCell, Lebanon, NH, USA) intraperitoneally (i.p) at days 7, 10, and 14 after cell injection. Depletion of CD8^+^, CD4^+^_,_ or NK^+^ cells was achieved by i.p. inoculation of 100 μg of antimouse CD8α (clone 2.43; BioXCell), CD4 (clone GK1.5; BioXCell), or NK1.1 (clone PK136; BioXCell), at days 6, 11, 14, 18, 21, and 28 after cell injection. Tumors were measured periodically using a digital caliper (DIN862, Ref 112-G, SESA Tools, Hernani, Spain), and tumor volume was calculated by the formula: Volume = π/6 × length × width^2^.

For liver colonization experiments, a murine model of intrasplenic cell inoculation was used for liver colonization experiments, as previously described^21^. The development of tumors was monitored in vivo using micro-PET (mPET). Total metabolic tumor volume (MTV) was measured at days 7, 14, and 21 post-injection. Mice were anesthetized and euthanized by cervical dislocation when tumor diameters reached 17 mm or when they appeared to be in distress.

### 4.10. ^18^F-FDG mPET Study

mPET studies were performed weekly by injecting the radiotracer 18-fluorodeoxyglucose (^18^F-FDG; 18.62 MBq ± 2 in 80–100 mL) via tail vein, as previously published [21]. PMOD software (PMOD Technologies Ltd., Zurich, Switzerland) was used to analyze the data. Images were expressed in standardized uptake value (SUV) units using the formula SUV = [tissue activity concentration (Bq/cm^3^)/injected dose (Bq)] × body weight (g).

After a qualitative assessment of the images, tumor foci were detected and volumes of interest were drawn, including the entire tumors. Then, a semi-automatic segmentation was performed to contour the tumors; this included voxels with a value greater than 50% of the maximum value of the tumor. Finally, metabolic tumor volume (MTV) of each lesion was calculated as the volume in cubic centimeters (cm^3^). Total MTV of each animal corresponds to the sum of the MTV values for all lesions.

### 4.11. Immunohistochemistry

Tumors were harvested and fixed in formaldehyde (3.7% to 4% (pH 7); Panreac, Castellar del Vallès, Spain) for 24 h. The relevant parts were embedded in paraffin, and sections were produced for hematoxylin-eosin staining (HE). Immunohistochemistry (IHC) was done as previously described^21^ using the antibody against Id1 (BCH-1/37-2; 1:1500, Biocheck). IHC to study the expression of CD3 (RM9107; 1:300, Neomarkers, Portsmouth, NH, USA), CD8 (#98941; 1:400, Cell Signaling, Danvers, MA, USA), CD4 (#ab183685; 1:1000, Abcam), and FOXP3 (FJK-16s; 1:200, Thermo) was performed by the Morphology Core Facility at CIMA. Slides were scanned with the Aperio Digital Scanner (Leica, Wetzlar, Germany) and analyzed with ImageJ software (NIH, Bethesda, MD, USA).

### 4.12. Multiplex Quantification of CD3/CD4/CD8

The NEL810001KT Opal kit (Akoya, Marloborough, MA, USA) was used for multispectral immunophenotyping and multiplexed automatic quantification of CD3, CD4, and CD8 in tumors. Unspecific binding was blocked in Dako Antibody diluent (Dako, Glostrup, Denmark) for 10 minutes at RT. After antigen retrieval using a citrate buffer (pH = 6), samples were incubated with anti-CD8 (1:400, Cell Signaling) and then with secondary antibody and Opal-690 (1:100, Akoya) fluorochrome. Then, antigen retrieval was performed using EDTA (pH = 9) followed by anti-CD4 (1:1000, Abcam), secondary antibody, and Opal-570 (1:100, Akoya). Finally, antigen retrieval with citrate buffer (pH = 6) was applied before adding the anti-CD3 (1:75, Abcam), secondary antibody, and Opal-520 (1:100, Akoya). Ready-to-use secondary antibodies and Opals are provided by the kit. We used DAPI for nuclear staining and Diamond antifade medium (Life Technologies) for mounting the slides.

Sample scanning, spectral unmixing, and quantification of signals were conducted with the Vectra Polaris Automated Quantitative Pathology Imaging System (Akoya), using the Phenochart and InForm 2.4 software (Akoya). The number of positive cells belonging to each specific phenotype was given as number of cells/square micron.

### 4.13. Flow Cytometry Analyses

LLC, 393P, and Lacun3 cell lines infected with *pLKO-Sc* or *pLKO-Id1sh* were plated (2 × 10^5^ cells per well) in 6-well plates. Recombinant murine interferon gamma (IFN*-γ*) (500 U/mL, PeproTech, Marlborough, MA, USA) was added, and the cells were incubated for 24 hours. Cells without IFN-γ were also analyzed. After 24 hours, the cells were detached using Dulbecco’s Phosphate-Buffered Saline (DPBS, Gibco) with 1 mM ethylenediaminetetraacetic acid (EDTA; Sigma-Aldrich) (pH 7.4) and centrifuged for 5 minutes at 1,200 r.p.m. CD274 (PD-L1, B7-H1) monoclonal antibody (MIH5) (1:500, Invitrogen) in AutoMacs buffer (1× DPBS, 2 mM EDTA and 0.5% bovine serum albumin/BSA; Sigma-Aldrich) was added for 20 minutes at room temperature to stain the cells. Cells were acquired in FACSCanto II Cytometer (BD Biosciences) and analyzed using FlowJo software v9.3.

Tumors were harvested and mechanically disrupted to obtain single-cell suspensions in DPBS (Lonza). Erythrocytes were lysed with ACK buffer (Gibco), and single-cell suspensions were treated with maleimide (#PK-PF670-3-01; PromoCell). Then, the cells were incubated with Fc block (#2.4G2; BD Pharmingen, San José, CA, USA) and stained with the following fluorochrome-conjugated antibodies: CD45 (#30-F11; Biolegend, San Diego, CA, USA), CD8a (#53-6.7; BD Bioscience), CD62L (#MEL-14; Biolegend), CD44 (#IM7; Biolegend), CD4 (#GK1.5; BD Bioscience), NK1.1 (#PK136; Biolegend), CD19 (#1D3; BD Bioscience), PD1 (#29F.1A12; Biolegend), Ly6C (#HK1.4; Biolegend), Ly6G (#1A8; Biolegend), CD11c (#HL3; BD Bioscience), CD11b (#M1/70; BD Bioscience), F4/80 (#BM8; Biolegend), MHC-C2 (#M5/114.15.2; Biolegend), GITR (#DTA-1; Biolegend), and LAG3 (#C9B7W; Biolegend). Intracellular fixation and permeabilization was performed using the Intracellular Fixation & Permeabilization Buffer Set (eBioscience, Waltham, MA, USA) according to the manufacturer’s instructions, and cells were stained with fluorochrome-conjugated antibodies against CD206 (#C068C2; Biolegend) and GzB (GB11; Biolegend). Samples were acquired using CytoFLEX flow cytometer (Beckman Coulter, Chaska, MN, USA). Data were analyzed using CytExpert software (Beckman Coulter). Gating strategies are shown in Appendix A.

### 4.14. Statistical Analysis

Individual results and median per group are shown in dot-plots. Mann–Whitney U test was used for comparisons between the two groups. Comparisons between treatment strategies were performed using the Kruskal-Wallis test with the Mann-Whitney U test as the *post hoc* test. Survival curves were generated using the Kaplan–Meier method, and differences were analyzed with the log rank test. For these analyses, survival times were defined as the number of days from the inoculation of the cells until mice were euthanized or died naturally. Significance in tumor volumes and survival tests were always calculated versus the group treated with the combined *Id1*/PD-1 blockade. *p* < 0.05 was considered statistically significant. Statistical analyses were performed using Prism software (GraphPad, San Diego, CA, USA).

## 5. Conclusions

In summary, these findings broaden our knowledge of PD-L1 regulation, introducing *Id1* as a novel relevant immune suppressive factor able to limit the efficacy of immune checkpoint inhibitors. Therefore, *Id1* targeting may contribute to significantly increasing the efficacy of anti-PD-1 ICIs and may help to overcome treatment resistance by significantly enhancing tumor infiltration by immune cells.

## Figures and Tables

**Figure 1 cancers-12-03169-f001:**
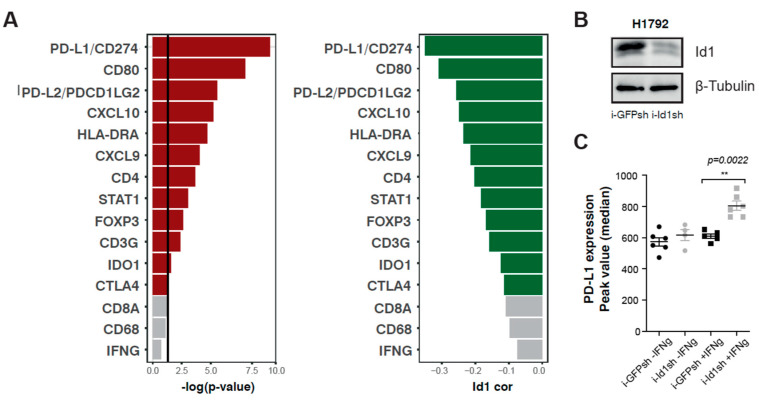
*Id1* expression inversely correlates with the mRNA expression levels of several markers related to immune response. (**A**) Pearson correlation coefficients (green) and *p*-values (red) between *Id1* mRNA expression levels and different genes associated with the immune system in LUAD patients. (**B**) Western blot for detection of *Id1* protein in human H1792 cells infected with doxycycline-inducible shRNA lentiviral particles that target *Id1*, in comparison with the same cell lines infected with a control i-GFPsh. (**C**) Flow cytometry analysis showed a statistically significant increase in PD-L1 expression after exposure to IFN-γ for 24 hours when human H1792 tumor cells had *Id1* inhibited (Median of peak value: H1792 +IFN-γ: i-GFPsh 610.0 [579.5–641.5], i-Id1sh 790.5 [734.0–874.5], *p* = 0.0022). The data are reported as the median with the interquartile range. ** *p* < 0.01.

**Figure 2 cancers-12-03169-f002:**
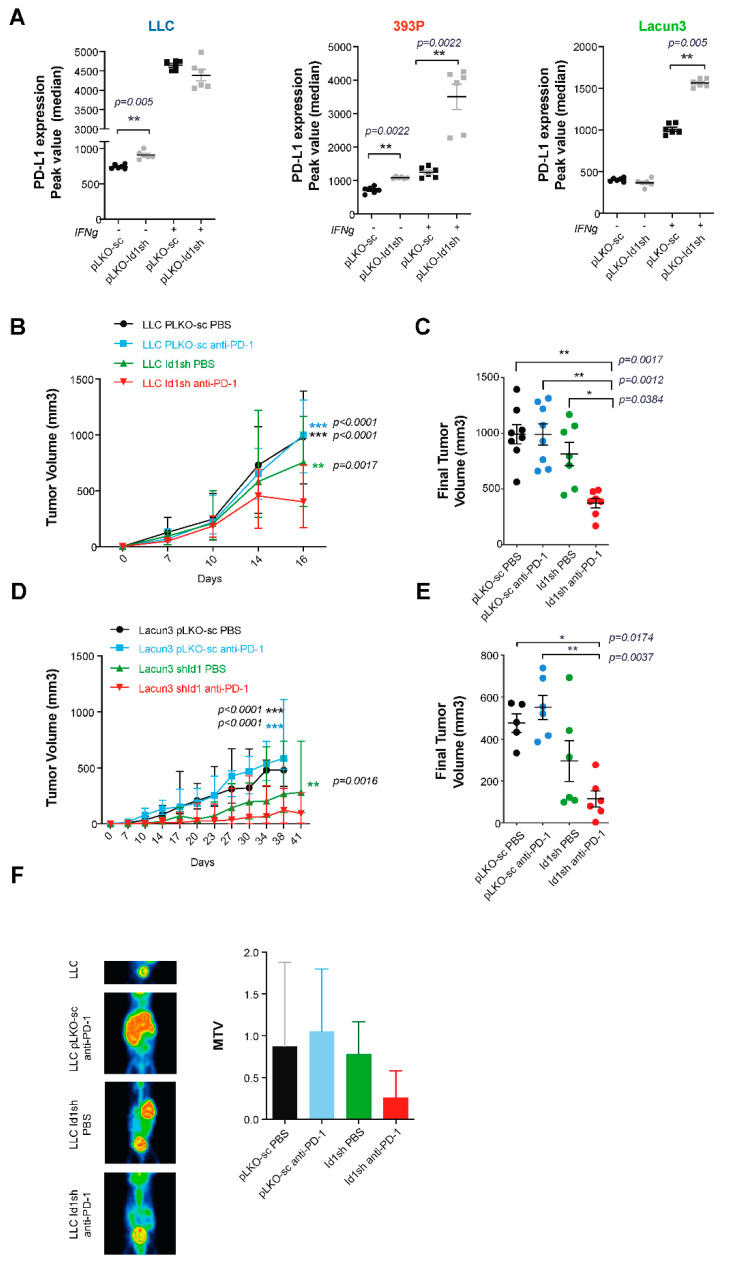
Silencing *Id1* gene in murine LUAD cell lines induce PD-L1 expression in vitro and potentiates the antitumor activity of PD-1 blockade in a mouse model. (**A**) Flow cytometry analysis showed a statistically significant increase in PD-L1 expression when LLC, 393P and Lacun3 tumor cells had *Id1* inhibited, in some cases after exposure to IFN-γ for 24 hours (Median of peak value: LLC -IFN-γ: pLKO-sc 737 [717–778], *Id1*sh 895 [875.3–953.5], *p* = 0.005); 393P -IFN-γ: pLKO-sc 729 [624.8–791.3], *Id1*sh 1076 [1059–1115], *p* = 0.0022; 393P +IFN-γ: pLKO-sc 1252 [1087–1394], *Id1*sh 3990 [2336 – 4184], *p* = 0.0022; Lacun3 +IFN-γ: pLKO-sc 989 [962.5–1085], *Id1*sh 1556 [1523–1617], *p* = 0.005). Results are represented as the average of three independent experiments. (**B**) Inhibition of *Id1* in LLC cells along with anti-PD-1 treatment impairs tumor growth in vivo in a syngeneic model of LUAD using C57BL/6J mice. (**C**) Final tumor volumes of the same four mice groups as in B at day 16 post-tumor injection (*Id1*sh/anti-PD-1: 388.8 [277.1–476.3] mm^3^; *Id1*sh/DPBS: 795.5 [499.4–1067] mm^3^, *p* = 0.0384; pLKOsc/anti-PD-1: 1002 [698.8–1267] mm^3^
*p* = 0.0012; pLKOsc/DPBS: 982.6 [869.4–1164] mm^3^, *p* = 0.0017). (**D**) Inhibition of *Id1* in Lacun3 cells along with anti-PD-1 treatment impairs tumor growth in vivo in a syngeneic model of LUAD using Balb/c mice. (**E**) Final tumor volumes of the same four mice groups as in D at days 38 (mice injected with Lacun3 pLKO-sc) and 41 (mice injected with *Id1*sh-Lacun3) post-tumor injection (*Id1*sh/anti-PD-1: 93.26 [43.82–192.4] mm^3^; *Id1*sh/DPBS: 218.1 [105.8–504.4] mm^3^, n.s.; pLKO/anti-PD-1: 537.8 [410.2–701.8] mm^3^, *p* = 0.0037; pLKO/DPBS: 481 [383.9–568.4] mm^3^, *p* = 0.0174). (**F**) Representative mPET images of ^18^F-FDG uptake in LLC (pLKO-sc or *Id1*sh) tumor-bearing C57BL/6J mice 3 weeks after intrasplenic tumor cell inoculation. Total metabolic tumor volumes (MTV) were quantified for each group and plotted (Id1sh/anti-PD-1: 0.26 [0–0.88]; *Id1*sh/DPBS: 0.69 [0.24–1.08]; pLKO/anti-PD-1: 0.57 [0.28–1.54]; pLKO/DPBS: 0.46 [0.23–1.93], *p* > 0.05). The data are reported as the median with the interquartile range. * *p* < 0.05, ** *p* < 0.01, *** *p* < 0.001.

**Figure 3 cancers-12-03169-f003:**
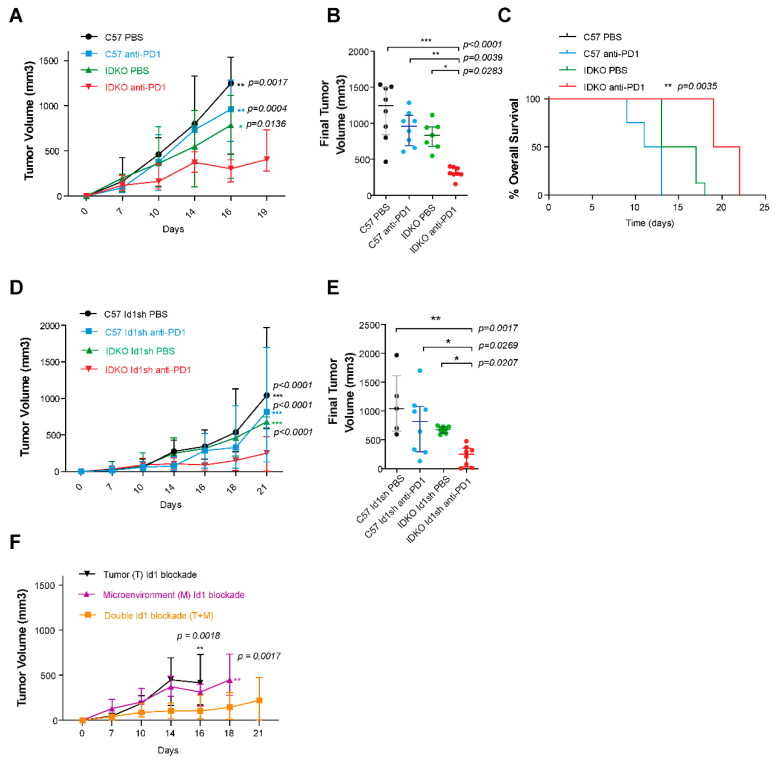
Effect of combined PD-1 and Id1 blockade in both tumor microenvironment (IDKO mice) and tumor microenvironment along with tumor cells (LLC) in a LUAD syngeneic model. (**A**) Tumor growth LLC parental cells injected in Id1+/+ (C57BL/6J) or Id1-/- (IDKO) mice. (**B**) Final tumor volumes of the same four mice groups as in A at days 16 and 19 (IDKO mice injected treated with anti-PD-1 therapy) post-tumor injection (Id1-/-/anti-PD-1: 303.1 [291–387.8] mm^3^; Id1-/-/DPBS: 834.6 [678.1–961.1] mm^3^, *p* = 0.0283; Id1+/+/anti-PD-1: 961.1 [686.7–1112 ] mm^3^, *p* = 0.0039; Id1+/+/DPBS: 1248 [839.3–1502] mm^3^, *p* < 0.0001). (**C**) Kaplan-Meier plot of the OS of mice in A (median survival per group in days: Id1-/-/anti-PD-1: 20.5; Id1-/-/DPBS: 15; Id1+/+/anti-PD-1: 12; Id1+/+/DPBS: 11; *p* = 0.0035). (**D**) Combined blockade of PD-1 and Id1 silencing both in the tumor microenvironment (IDKO mice) and LLC cells significantly decreases tumor proliferation. (**E**) Final tumor volumes of the same four mouse groups as in D at day 21 post-tumor injection (Id1-/-/anti-PD-1: 250.7 [30.34–357.4] mm^3^; Id1-/-/DPBS: 679.8 [617.4–728.2] mm^3^, *p* = 0.0207; Id1+/+/anti-PD-1: 817.5 [296.6–1077] mm^3^, *p* = 0.0269; Id1+/+/DPBS: 1043 [649–1613] mm^3^, *p* = 0.0017). *** *p* < 0.001. (**F**) Comparison of tumor growth of the three murine models injected with the LLC cell line (Tumor Id1 blockade (T), Microenvironment Id1 blockade (M) and double Id1 blockade (T + M)) and treated with anti-PD-1 therapy. The data are reported as the median with the interquartile range. * *p* < 0.05, ** *p* < 0.01.

**Figure 4 cancers-12-03169-f004:**
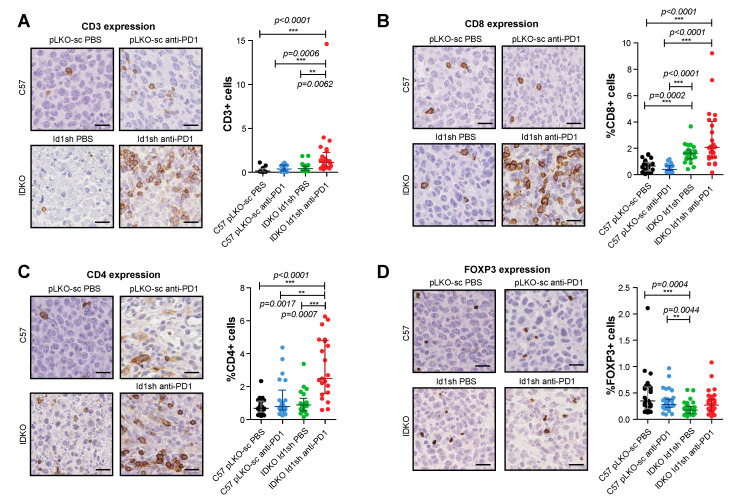
Combined inhibition of *Id1* and anti-PD-1 therapy promotes the tumor infiltration of different immune cell populations. (**A**) HC and quantification of CD3 expression in representative sections of tumors belonging to group of mice with extreme phenotypes [*Id1+/+*/DPBS (pLKO-sc LLC cells), *Id1+/+*/anti-PD-1 (pLKO-sc LLC cells), *Id1-/-/*DPBS (Id1sh LLC cells) and *Id1-/-/*anti-PD-1 (Id1sh LLC cells]. (**B**) IHC and quantification of CD8 expression in representative sections of the same tumors as in A. (**C**) IHC and quantification of CD4 expression in representative sections of the same tumors as in A. (**D**) IHC and quantification of FOXP3 expression in representative sections of the same tumors as in A. The data are reported as the median with the interquartile range or mean ± SD. ** *p* < 0.01, *** *p* < 0.001, n.s. (not significant). Scale bar, 30 µm.

**Figure 5 cancers-12-03169-f005:**
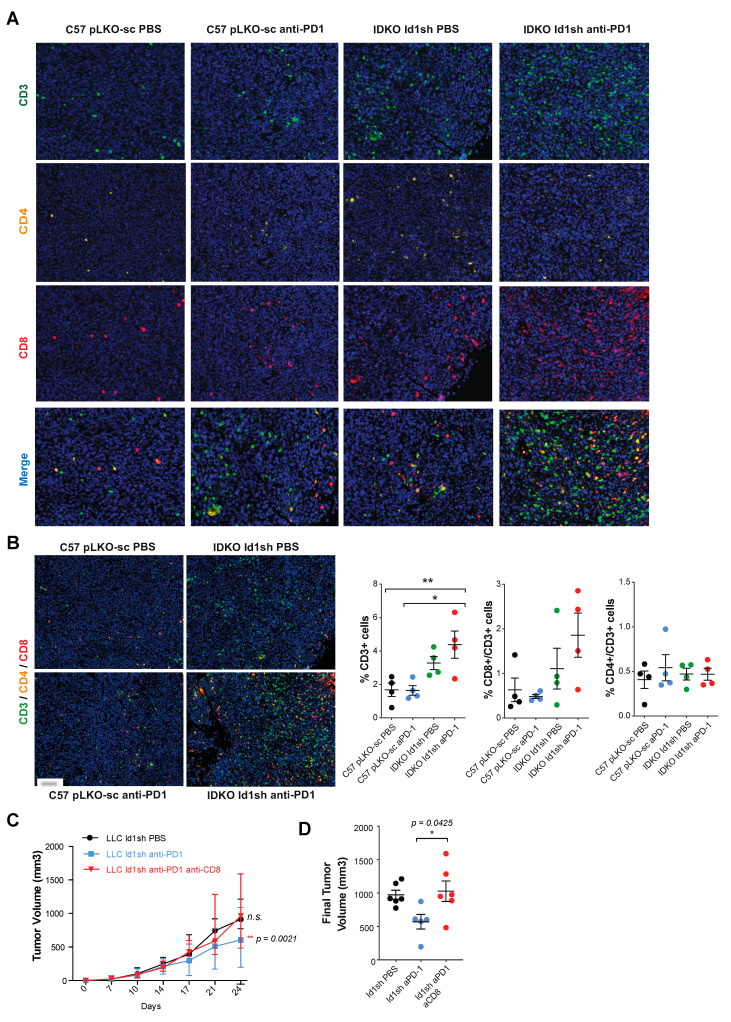
CD8^+^ T cells inflammatory infiltrate may mediate the anti-tumor activity observed after the Id1 and PD-1 blockade. (**A**) Quantitative multiplex IHC (CD3, CD4, CD8 and multiplex) in representative sections of tumors belonging to group of mice with extreme phenotypes [Id1+/+/DPBS (pLKO-sc LLC), Id1+/+/anti-PD-1 (pLKO-sc LLC), Id1-/-/DPBS (Id1sh LLC) and Id1-/-/anti-PD-1 (Id1sh LLC); four mice per group]. (**B**) Representative merged images derived from the multiplex quantification of CD3/CD4/CD8 in the same tumors as in A confirm the higher TIL when tumors lack Id1 and are treated with anti-PD-1 therapy. Quantification of the percentage of CD3, CD8/CD3, and CD4/CD3 positive cells. (**C**) In vivo tumor growth of Id1sh-LLC cells in C57BL/6J mice treated with CD8^+^ T cells depleting antibodies. Tumors were measured on days 7, 10, 14, 17, 21, and 24. (**D**) Final tumor volumes of the same three mice groups as in C at day 24 post-tumor injection (Id1sh/anti-PD-1/anti-CD8: 962.1 [786.0–1360] mm^3^; Id1sh/DPBS: 914.5 [853.6–1160] mm^3^, n.s.; Id1sh/anti-PD-1: 602.3 [386.4–740.5] mm^3^, *p* = 0.0425). The data are reported as the median with the interquartile range or mean ± SD. * *p* < 0.05, ** *p* < 0.01, *** *p* < 0.001, n.s. (not significant). Scale bar, 30 µm.

**Figure 6 cancers-12-03169-f006:**
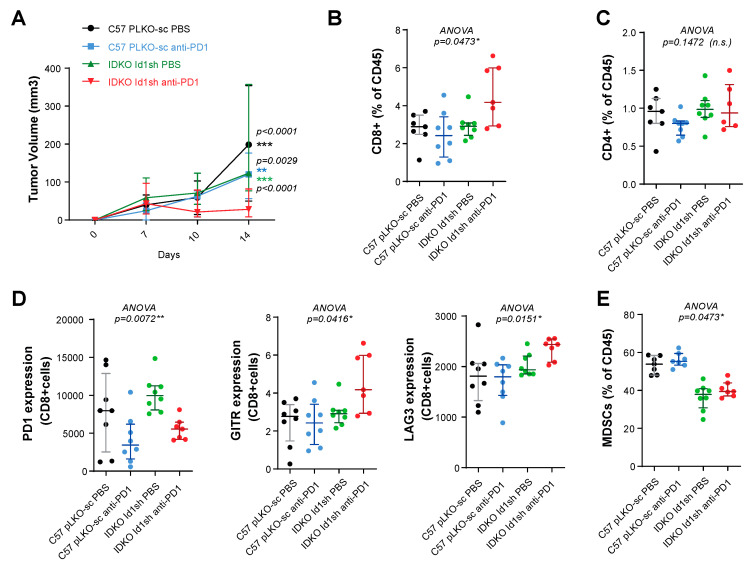
The antitumor activity observed after the Id1 and PD-1 blockade is mediated by CD8^+^ T cells. (**A**) Tumor growth of LLC (pLKO-sc or Id1sh) cells in mice (C57BL/6J or IDKO) treated with DPBS or anti-PD-1 for endpoint flow cytometry analysis (final tumor volume per group: Id1-/-/anti-PD-1: 28.04 [21.42–65.61] mm^3^; Id1-/-/DPBS: 123.3 [82.24–227.2] mm^3^, *p* < 0.0001; Id1+/+/anti-PD-1: 120.3 [70.97–148.1] mm^3^, *p* = 0.0029; Id1+/+/DPBS: 198.4 [122.6–223.8] mm^3^, *p* < 0.0001). (**B**) Flow-cytometric analysis of CD8^+^ T cells in tumors harvested at day 14. (**C**) Flow-cytometric analysis of CD4^+^ T cells in tumors harvested at day 14. (**D**) Flow-cytometric analysis of PD-1, GITR and LAG3 markers (expressed as mean fluorescence intensity on CD8^+^ T cells) in LLC tumor grown the same mice as in A. (**E**) Flow-cytometric analysis of MDSCs cells in tumors harvested at day 14. The data are reported as the median with the interquartile range. * *p* < 0.05, ** *p* < 0.01, *** *p* < 0.001, n.s. (not significant).

**Table 1 cancers-12-03169-t001:** *Id1* expression correlation with immune response markers in the TCGA data set.

**Gene ID**	**IFNg Signature *(Seiwert et al. 2016)***											
	***CXCL10***		***HLA-DRA***		***CXCL9***		***STAT1***		***IDO1***		***IFNg***	
	**cor**	***p*** **-value**	**cor**	***p*** **-value**	**cor**	***p*** **-value**	**cor**	***p*** **-value**	**cor**	***p*** **-value**	**cor**	***p*** **-value**
***Id1***	−0.25	0.015	−0.23	0.013	−0.21	0.066	−0.18	0.176	−0.12	0.364	−0.074	0.713
**Gene ID**	**Markers of Immune Cell Populations**											
	***CD4***		***HLA-DRA***		***CD4***		***STAT1***		***CD4***		***IFNg***	
	**cor**	***p*** **-value**	**cor**	***p*** **-value**	**cor**	***p*** **-value**	**cor**	***p*** **-value**	**cor**	***p*** **-value**	**cor**	***p*** **-value**
***Id1***	−0.20	0.042	−0.17	0.120	−0.16	0.217	−0.11	0.684	−0.10	0.408	−0.20	0.042
**Gene ID**	**Immune Checkpoints**											
	***CD274*** (***PD-L1***)		***HLA-DRA***		***CD274*** (***PD-L1***)		***STAT1***		***CD274*** (***PD-L1***)		***IFNg***	
	**cor**	***p*** **-value**	**cor**	***p*** **-value**	**cor**	***p*** **-value**	**cor**	***p*** **-value**	**cor**	***p*** **-value**	**cor**	***p*** **-value**
***Id1***	−0.35	0.0001	−0.31	0.0003	−0.26	0.007	−0.113	0.568	−0.12	0.364	−0.074	0.713

**Table 2 cancers-12-03169-t002:** Immunohistochemical analysis of markers of tumor infiltrating lymphocytes from *Id1+/+*/DPBS (pLKO-sc LLC), *Id1+/+*/anti-PD1 (pLKO-sc LLC), *Id1-/-/*DPBS (*Id1*sh LLC), and *Id1-/-/*anti-PD1 (*Id1*sh LLC) tumors.

IHC Marker	C57 pLKO-sc PBS		C57 pLKO-sc anti-PD1		IDKO Id1sh PBS		IDKO Id1sh anti-PD1	
	Median	Interquartile Range	Median	Interquartile Range	Median	Interquartile Range	Median	Interquartile Range
***CD3***	0.158	0.089–0.388	0.368	0.147–0.840	0.413	0.205–0.711	1.136	0.626–2.267
***CD8***	0.655	0.283–0.949	0.403	0.320–0.674	1.638	1.202–1.870	2.072	1.321–4.045
***CD4***	0.680	0.448–1.128	0.788	0.588–1.783	0.887	0.539–1.297	2.490	1.580–4.810
***FOXP3***	0.35	0.2–0.615	0.280	0.230–0.385	0.175	0.113–0.248	0.270	0.155–0.385

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
