# Peer review of "Id1 and PD-1 Combined Blockade Impairs Tumor Growth and Survival of *KRAS*-mutant Lung Cancer by Stimulating PD-L1 Expression and Tumor Infiltrating CD8^+^ T Cells"

_cancers, 2020, doi:10.3390/cancers12113169_

Round 1

Reviewer 1 Report

The work is well written, articulated and methodologically correct.

Authors have long experience on ID-1 research and their previous publications are in line with the objective of this study. In the present study, the authors provide a concept of proof on the role of inhibiting Id-1 to potentiate the response to anti-PD-1 inhibition both in vitro and in vivo, opening the way to explore the combination of Id-1 and PD-1 inhibitors with the aim to improve clinical outcomes of patients with lung adenocarcinoma.

Minor comments: minor spell check is suggested

Line 33: typos to be corrected

Lines 60-63: the sentence is too long. you should make it easier

Line 112: typo to be corrected. Did the authors refer to table 1?

Author Response

RESPONSE TO REVIEWER 1 COMMENTS

The work is well written, articulated and methodologically correct.

Authors have long experience on ID-1 research and their previous publications are in line with the objective of this study. In the present study, the authors provide a concept of proof on the role of inhibiting Id-1 to potentiate the response to anti-PD-1 inhibition both in vitro and in vivo, opening the way to explore the combination of Id-1 and PD-1 inhibitors with the aim to improve clinical outcomes of patients with lung adenocarcinoma.

Point 1: Minor comments: minor spell check is suggested. Line 33: typos to be corrected. Lines 60-63: the sentence is too long. you should make it easier

Response 1: We appreciate these comments; spelling errors have been corrected. We have corrected the typo in line 33: “immunosuppression”. The sentence in line 60-63 has been re-written to “The discovery of actionable driver genomic alterations that confer sensitivity to targeted therapies and the ability of immune checkpoint inhibitors (ICIs) to harness an anti-tumor immune response have revolutionized therapy for advanced NSCLC. The implementation of science-based precision medicine has resulted in a significant improvement in both patients’ life expectancy and quality of life [3].

Point 2: Line 112: typo to be corrected. Did the authors refer to table 1?

Response 2: We have clarified this sentence in the manuscript:P-value and the magnitude of the correlation are showed in Table 1 and ranked in Fig.1A”.

Reviewer 2 Report

The authors first found in silico that ID1 expression inversely correlates with mRNA levels of several immune response markers related to INFg signature, immune cell population, and immune checkpoint, and then showed that Id1 and PD-1 combined blockade impairs tumor growth and survival of KRAS-mutant lung cancer by stimulating PD-L1 expression and tumor infiltrating CD8+ T cells by using in vitro and in vivo KRAS-mutant lung adenocarcinoma models and also by co-culture assay using CD8+ T cells and Id1-knock down tumor cells.

Comments

1. On line 626 on page 19, supplementary Fig. 1 should be the last supplemental Fig as follows: Suppl. Fig. 1 to suppl. Fig. 5, suppl. Fig. 2 to suppl. Fig. 1, suppl. Fig. 3 to suppl. Fig. 2, suppl. Fig. 4 to suppl. Fig. 3, and suppl. Fig. 5 to suppl. Fig. 4.

2. In in silico search, the author found that Id1 expression inversely correlates with mRNA levels of ligands (CXCL9 and 10) for TIL-expressing CXCR3. Does Id1 blockade increase mRNA levels of both CXCL9 and 10? Please discus them.

3. How about ligands (CCL3, 4, 5, and 8) for TIL-expressing CCR5?

Author Response

RESPONSE TO REVIEWER 2 COMMENTS

The authors first found in silico that ID1 expression inversely correlates with mRNA levels of several immune response markers related to INFg signature, immune cell population, and immune checkpoint, and then showed that Id1 and PD-1 combined blockade impairs tumor growth and survival of KRAS-mutant lung cancer by stimulating PD-L1 expression and tumor infiltrating CD8+ T cells by using in vitro and in vivo KRAS-mutant lung adenocarcinoma models and also by co-culture assay using CD8+ T cells and Id1-knock down tumor cells.

Point 1: On line 626 on page 19, supplementary Fig. 1 should be the last supplemental Fig as follows: Suppl. Fig. 1 to suppl. Fig. 5, suppl. Fig. 2 to suppl. Fig. 1, suppl. Fig. 3 to suppl. Fig. 2, suppl. Fig. 4 to suppl. Fig. 3, and suppl. Fig. 5 to suppl. Fig. 4.

Response 1: We would like to thank the reviewers for this recommendation, we have changed the manuscript accordingly.

Point 2: In in silico search, the author found that Id1 expression inversely correlates with mRNA levels of ligands (CXCL9 and 10) for TIL-expressing CXCR3. Does Id1 blockade increase mRNA levels of both CXCL9 and 10? Please discus them.

Response 2: We appreciate this comment. We found an inverse correlation between CXLC9/CXCL10 and Id1. Therefore, an increase in mRNA levels of CXCL9 and 10 is expected when Id1 is blocked. However, we have not measured CXCL9 and CXCL10 expression upon Id1 inhibition and we cannot further assure this statement.

Point 3: How about ligands (CCL3, 4, 5, and 8) for TIL-expressing CCR5?

Response 3: We have performed this analysis and found an inverse and statistically significant correlation for CCL4 and CCL8. However, this correlation is extremely weak (cor: -0.14 and -0.17 respectively). The result is attached hereby.

Please see the figure at the attachment.

Reviewer 3 Report

Dear Authors,

you have presented very interesting findings in your article. However, I have some minor notes/comments:

Figure 1 is in part not readable (at least in my copy)

Figure 2 (E): why did you choose d38 and d41 for the "last" read out? at these timepoints you have lost already several animals (due to complications? or tumor size?), in my opinion especially in the group of pLKO-sc PBS the reduction of animals by 50% biases your results.

Line 131-138: you mention the KRAS-dependent stratification of your cohort including the subsequent analysis. what are the results if one would stratify for PDL1 status and Id1 status, respectively?

Line 209ff: you use a subcutaneous xenograft model for your study; is data available also from orthotopic models? (I guess not since you reference it in your discussion; should I be wrong, please include it!) I understand that a xenograft model is a standardized and well-understood/used model, however, several researchers have already shown that the impact on/the influence of immune cells differs immensely from spontanous or orthotopic models.

I find it very interesting that the selective depletion of CD8+ cells in the combined inhibition of PDL1 and Id1 is essential for antitumoral acitivity. I am curious what a pharmacologic inhibition of Id1 (as you have mentioned) will show.

Author Response

RESPONSE TO REVIEWER 3 COMMENTS

Dear Authors, you have presented very interesting findings in your article. However, I have some minor notes/comments:

Point 1: Figure 1 is in part not readable (at least in my copy)

Response 1: We appreciate this comment. We have noticed that figure 1 is readable in the word version of the manuscript, but it is partially unreadable in the pdf version. We attached again a corrected version of this figure in the manuscript.  

Point 2: Figure 2 (E): why did you choose d38 and d41 for the "last" read out? at these timepoints you have lost already several animals (due to complications? or tumor size?), in my opinion especially in the group of pLKO-sc PBS the reduction of animals by 50% biases your results.

Response 2: In the in vivo assay using Lacun3 cell line, 5-6 mice were used per group. Sequential tumor volume is represented in figure 2D. Last read out differs according to group of treatment since mice in the Lacun3 shId1 PBS and Lacun3 shId1 antiPD-1 presented less tumor volume and better functional status. Mice in the other groups were sacrificed at day 38, as welfare of animals used for research, while the aforementioned were sacrificed at day 41. Final tumor volume for all mice included in the experiment is represented in figure 2E, there is no reduction of animals by 50%, so our results are not biased.   

Point 3: Line 131-138: you mention the KRAS-dependent stratification of your cohort including the subsequent analysis. what are the results if one would stratify for PDL1 status and Id1 status, respectively?

Response 3: We appreciate this comment. However, mutations in PD-L1 are extremely rare and Id1 mutations have never been described.

Point 4: Line 209ff: you use a subcutaneous xenograft model for your study; is data available also from orthotopic models? (I guess not since you reference it in your discussion; should I be wrong, please include it!) I understand that a xenograft model is a standardized and well-understood/used model, however, several researchers have already shown that the impact on/the influence of immune cells differs immensely from spontanous or orthotopic models.

Response 4: As the reviewer points out, we have not used orthotopic models in our study. Even if orthotopic models encounter organ-specific tumor microenvironment, subcutenous xenograft models are known to be an optimal model to study the tumor biology.

Point 5: I find it very interesting that the selective depletion of CD8+ cells in the combined inhibition of PDL1 and Id1 is essential for antitumoral acitivity. I am curious what a pharmacologic inhibition of Id1 (as you have mentioned) will show.

Response 5: We appreciate this enthusiastic comment and are currently working on the pharmacologic inhibition of Id1.